# An optogenetic method for the controlled release of single molecules

Purba Kashyap[1], Sara Bertelli[2,8], Fakun Cao[3,8], Yulia Kostritskaia [4,8], Fenja Blank [1,8], Niranjan A. Srikanth [1,3], Claire Schlack-Leigers[1], Roberto Saleppico[5], Dolf Bierhuizen [1], Xiaocen Lu [6], Walter Nickel[5], Robert E. Campbell [6,7], Andrew J. R. Plested [2], Tobias Stauber [4], Marcus J. Taylor [3] & Helge Ewers[1] ✉

We developed a system for optogenetic release of single molecules in cells. We confined soluble and transmembrane proteins to the Golgi apparatus via a photocleavable protein and released them by short pulses of light. Our method allows for a light dose-dependent delivery of functional proteins to the cytosol and plasma membrane in amounts compatible with single-molecule imaging, greatly simplifying access to single-molecule microscopy of any protein in live cells. We were able to reconstitute ion conductance by delivering BK and LRRC8/volume-regulated anion channels to the plasma membrane. Finally we were able to induce NF-kB signaling in T lymphoblasts stimulated by interleukin-1 by controlled release of a signaling protein that had been knocked out. We observed light-induced formation of functional inflammatory signaling complexes that triggered phosphorylation of the inhibitor of nuclear factor kappa-B kinase only in activated cells. We thus developed an optogenetic method for the reconstitution and investigation of cellular function at the single-molecule level.

Single-molecule fluorescence microscopy is a powerful technique in the investigation of protein function. Many fundamental cellular processes, including stepping of motor molecules[1], DNA replication[2], transcription[3,4] and translation[5], or the stoichiometry[6] and mechanism of action[7] of membrane receptors, have been elucidated using single-molecule imaging. However, this method requires a sparse population of labeled molecules to avoid signal overlap. At the same time, it is essential that no unlabeled endogenous background population interferes with measurements, which is challenging in practice. While genome engineering has alleviated the problem of quantitative and endogenous labeling, the controlled delivery of single molecules remains an unsolved challenge because existing approaches are either (1) too leaky for control at the single-molecule level because they are based on noncovalent attachment or (2) require complex expression systems such as messenger RNA injection[8].

Here we developed a technique for the optogenetically controlled release of functional single molecules in live cells. We make use of a recently developed green-to-red photoconvertible protein called PhoCl[9] that is photocleaved, splitting in two following activation in near-ultraviolet (near-UV) light (Fig. 1a). In PhoCl, the fluorophore encoded by the amino acid sequence breaks following UV illumination, severing the peptide chain. Because PhoCl is circularly permutated, the fluorophore is located towards the end of the coding sequence.

[1]Institut für Chemie und Biochemie, Freie Universität Berlin, Berlin, Germany. [2]Humboldt-Universität zu Berlin and Leibniz-Forschungsinstitut für Molekulare Pharmakologie, Berlin, Germany. [3]Max-Planck-Institute for Infection Biology, Berlin, Germany. [4]Institute for Molecular Medicine, MSH Medical School Hamburg, Hamburg, Germany. [5]Heidelberg University Biochemistry Center, Heidelberg, Germany. [6]Department of Chemistry, University of Alberta, Edmonton, Alberta, Canada. [7]Department of Chemistry, The University of Tokyo, Tokyo, Japan. [8]These authors contributed equally: Sara Bertelli, Fakun Cao, Yulia Kostritskaia, Fenja Blank. ✉e-mail: helge.ewers@fu-berlin.de

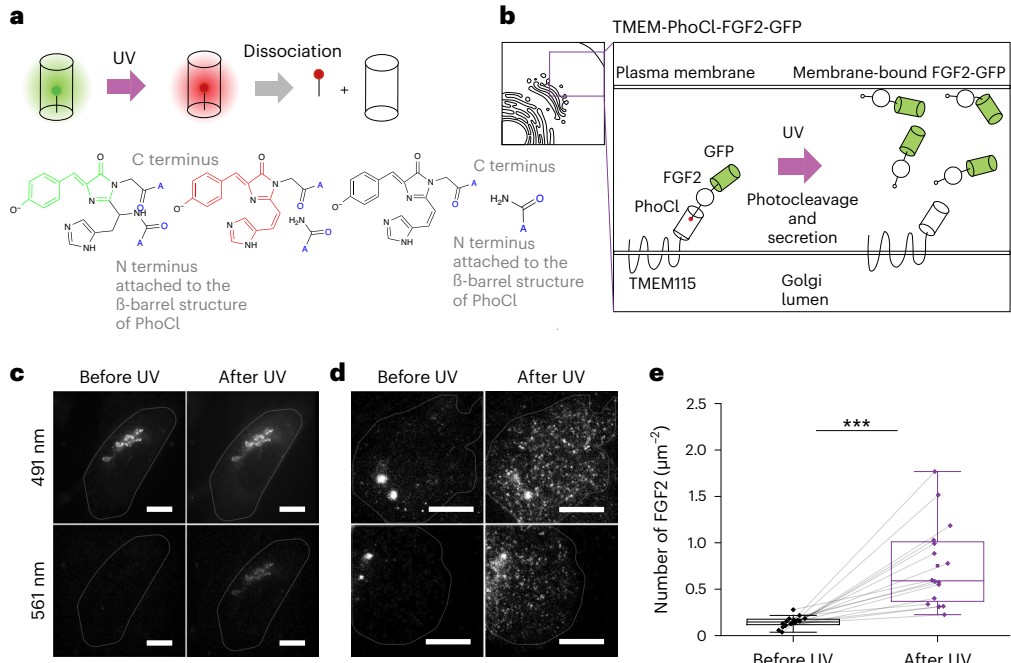

**Fig. 1 | PhoCl-based uncaging of cytosolic protein FGF2-GFP. a**, Schematic of light-induced photoconversion and dissociation of PhoCl. **b**, Schematic of TMEM-PhoCl-FGF2-GFP construct in CHO-K1 cells. **c**, Images of a cell expressing TMEM-PhoCl-FGF2-GFP in 491- and 561-nm channels before and after UV illumination. **d**, TIRF images of cells expressing TMEM-PhoCl-FGF2-GFP before and after UV illumination. **e**, Quantification of membrane-bound FGF2-GFP before ($0.15 \pm 0.06\ \mu m^{-2}$) and after UV illumination ($0.75 \pm 0.4\ \mu m^{-2}$) ($n = 2$ and $n = 16$, respectively). Significance was tested using two-tailed paired sample sign testing (***$P = 10^{-5}$). Data presented as mean ± s.d. $N$ denotes the number of biological replicates, $n$ the number of cells. Center line in the box plot represents the median, the box represents 25–75% of the data, whiskers represent $1.5 \times$ interquartile range and the square box among data points represents the mean. **c,d**, Scale bars, 10 μm.

Following breakage, the C terminus thus dissociates from the protein barrel structure.

Here we sequestered target proteins to the Golgi apparatus by fusing their coding sequence via PhoCl to Golgi-resident proteins. UV-induced cleavage of PhoCl would thus result in the barrel structure of PhoCl remaining at the Golgi apparatus whereas the C-terminal peptide and fused target molecule would be released. This system allowed us to release functional target molecules at levels compatible with single-molecule imaging in a light-controlled manner. At the same time, fusion to a Golgi-resident protein allowed for covalent attachment to the Golgi apparatus, resulting in low leakiness. We demonstrate the function of our system with the examples of a cytosolic protein, a single-spanning transmembrane protein, multisubunit ion channels and a kinase within a signaling complex. Our system facilitated controlled, single-molecule imaging at the plasma membrane and the surface delivery of single-transmembrane proteins. Furthermore, we were able to reconstitute specific ion channel conductance and restore a signaling pathway in knockout cells by releasing an essential kinase and subunit of the Myddosome signaling complex. We provide a new experimental paradigm that will allow for both quantitative, single-molecule imaging and optogenetic control over membrane protein function.

## Results

### Controlled release of single-molecule cytosolic proteins

We first asked whether a soluble molecule could be confined to a specific location in the cell and released following photocleavage of PhoCl. To do so we made use of FGF2, a cytosolic protein secreted via an unconventional pathway through the plasma membrane[10,11]. Before penetrating the plasma membrane, FGF2 can be readily visualized by single-molecule total internal reflection fluorescence (TIRF) microscopy at the inner membrane leaflet and, following

secretion, detected from outside of the cell[9]. To sequester FGF2 outside the evanescent field of TIRF, we generated a fusion construct from FGF2, PhoCl and Golgi-resident transmembrane protein TMEM115 (Fig. 1b). When we expressed this construct we observed FGF2-green fluorescent protein (GFP) at the Golgi apparatus (Fig. 1c). When we then illuminated the sample with brief pulses of 405-nm light we detected the red fluorescence of PhoCl appearing at the Golgi apparatus, besides persistent green fluorescence, indicating breakage of the peptide chain backbone in a fraction of molecules (Fig. 1c). Consistent with this, quantification before and 10 min following activation showed an increase in overall cytosolic FGF2-GFP fluorescence (Supplementary Fig. 1). Soon after the UV pulse, highly mobile single FGF2-GFP spots appeared in the TIRF field (Supplementary Video 1 and Supplementary Fig. 2), suggesting recruitment to the inner plasma membrane leaflet (Fig. 1d,e). We concluded that FGF2-GFP sequestration to the Golgi apparatus, and its optogenetic release, were successful and yielded functional FGF2 that was recruited to the plasma membrane.

### Controlled release of single-molecule transmembrane proteins

Next we aimed to use our optogenetic method to release single-transmembrane proteins. To do so we used mScarlet-CD4, a generic, single-spanning plasma membrane protein. We created a construct containing the coding sequences of mScarlet-CD4, PhoCl and retrieval protein-1 (RER), a Golgi-resident protein (Fig. 2a). RER successfully anchored mScarlet-CD4 at the Golgi apparatus (Supplementary Fig. 3a). As before, UV illumination of PhoCl led to uncaging and trafficking of mScarlet-CD4 to the plasma membrane where multiple, laterally mobile, single-molecule spots appeared on TIRF microscopy (Fig. 2b and Supplementary Video 2). To ensure that only mScarlet-CD4 molecules properly inserted into the plasma membrane were

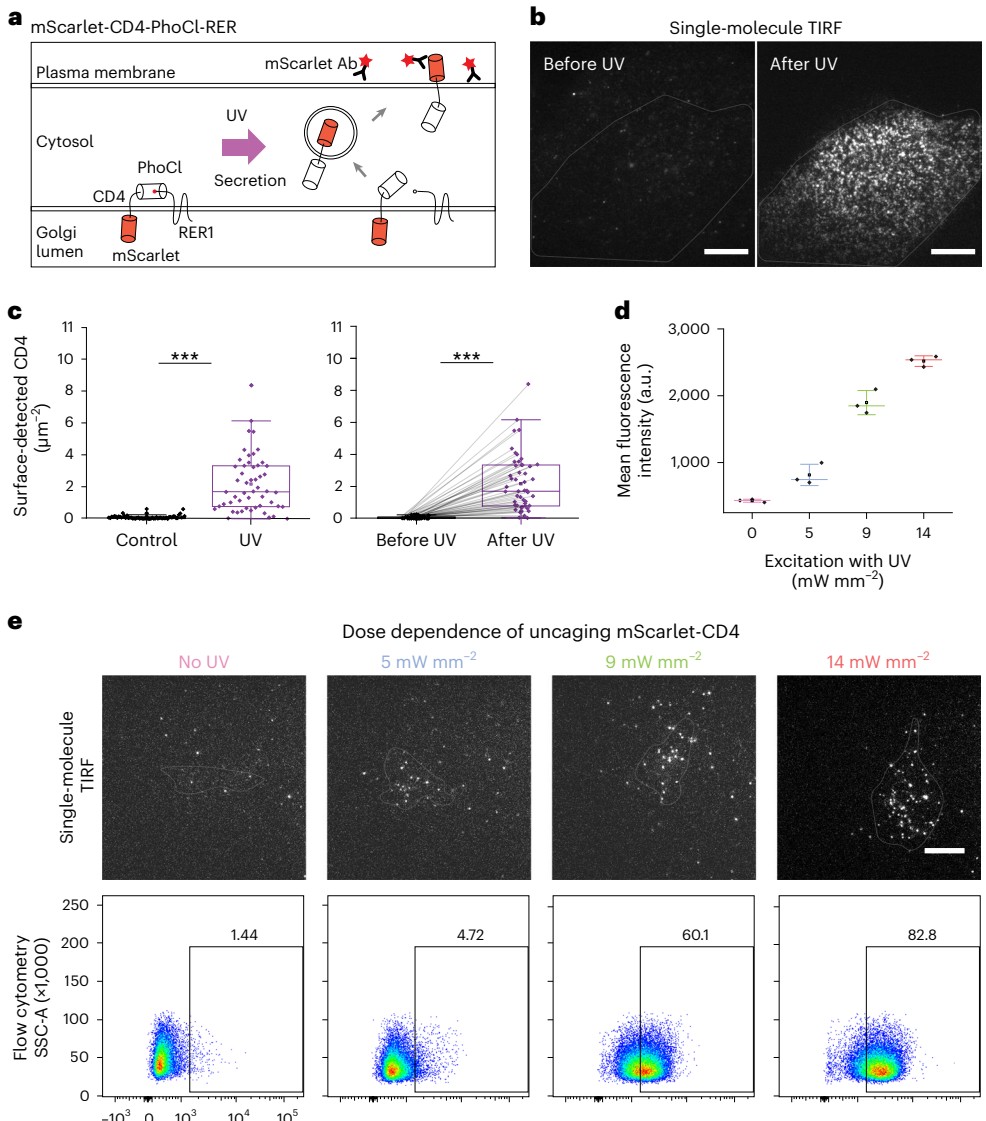

**Fig. 2 | PhoCl-based uncaging of transmembrane protein CD4. a**, Schematic of mScarlet-CD4-PhoCl-RER construct. Plasma membrane-localized mScarlet-CD4 is recognized by AF647-labeled antibodies in the medium. **b**, TIRF images of a CV-1 cell expressing mScarlet-CD4-PhoCl-RER with antibodies against mScarlet before and after UV illumination. **c**, Left: quantification of the number of anti-mScarlet antibodies bound to control ($0.1 \pm 0.1$ µm$^{-2}$, $n = 55$) and to UV-illuminated cells ($2.20 \pm 1.78$ µm$^{-2}$, $N = 11$, $n = 52$). Significance was tested using a two-tailed Mann–Whitney U-test (***$P = 10^{-16}$). Right: quantification of the number of anti-mScarlet antibodies bound to the same cells before ($0.07 \pm 0.07$ µm$^{-2}$) and after uncaging ($2.16 \pm 1.77$ µm$^{-2}$), $N = 11$, $n = 51$. Significance was tested using the two-tailed paired sample sign test (***$P = 10^{-9}$). **d**, Mean fluorescence intensity (arbitrary units, a.u.) of AF647-labeled mScarlet

antibodies bound to HeLa cells expressing mScarlet-CD4-PhoCl-RER following varying doses of UV illumination ($429 \pm 22.3$ (0 UV), $815 \pm 159.5$ (5 mW mm$^{-2}$), $1,897 \pm 180.8$ (9 mW mm$^{-2}$), $2,518 \pm 79.3$ (14 mW mm$^{-2}$)). Bars represent median $\pm$ s.d. and data points represent the means from independent flow cytometry experiments. Square box represents the mean. $N = 3$. **e**, TIRF images (top) and flow cytometry results (bottom) of HeLa cells expressing mScarlet-CD4-PhoCl-RER with antibodies against mScarlet following varying doses of UV illumination. $N = 3$. Data presented as mean $\pm$ s.d. Center line in the box plot represents the median, the box represents 25–75% of the data, whiskers represent $1.5 \times$ interquartile range and the square box among data points represents the mean. Scale bars, 10 µm.

quantified, we detected mScarlet-CD4 from the exoplasmic space. To do so we added Alexa Fluor 647 (AF647)-labeled antibodies to live cells and quantified these by TIRF microscopy before and after activation (Fig. 2c). When we modified laser intensity, we found that the level of plasma membrane-inserted molecules could be controlled by the strength of the uncaging pulse (Fig. 2d,e and Supplementary Fig. 4), suggesting that the amount of released molecules can be controlled by light intensity. We concluded that PhoCl in combination with RER as a Golgi anchor can be used for optogenetic control of the delivery of a transmembrane protein to the plasma membrane at levels compatible with single-molecule imaging in a quantitative manner.

## Controlled release of functional multisubunit ion channels
Next we aimed to reconstitute functional ion channels into the cell plasma membrane. To do so we made use of the BK-channel, a voltage-gated and calcium-sensing potassium channel[12] that is assembled from four identical subunits. When we fused the coding sequence of the BK-channel to TMEM115 via PhoCl (Fig. 3a), we found that it localized to the Golgi apparatus following transient expression (Supplementary Fig. 3b). When we then illuminated cells with 5-s pulses of 405-nm light (33 mW mm$^{-2}$), we found that surface-delivered BK-channels could be detected in TIRF microscopy via AF647-coupled antibodies added to the medium (Fig. 3b–d). By contrast, these were not detectable

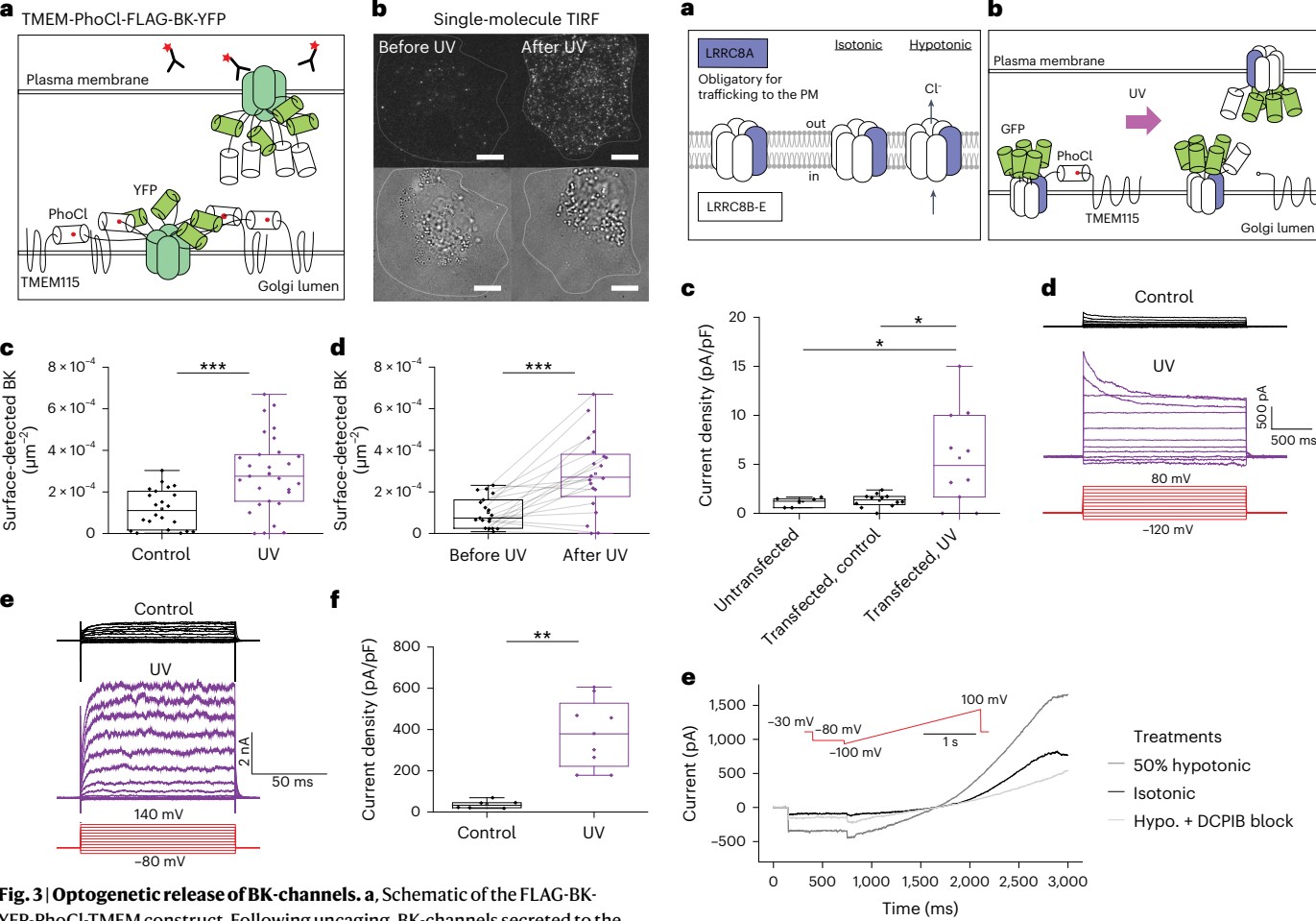

**Fig. 3 | Optogenetic release of BK-channels. a**, Schematic of the FLAG-BK-YFP-PhoCl-TMEM construct. Following uncaging, BK-channels secreted to the membrane were detected using antibodies in the medium. **b**, TIRF and brightfield images of a CV-1 cell expressing FLAG-BK-YFP-PhoCl-TMEM with antibodies against FLAG before and after UV illumination. **c**, Quantification of the number of anti-FLAG antibodies bound to control ($1.2 \times 10^{-4} \pm 9 \times 10^{-5}$, $n = 22$) and to UV-illuminated cells ($2.9 \times 10^{-4} \pm 1.8 \times 10^{-4}$, $n = 29$), $N = 5$. Significance was tested using the two-tailed Mann–Whitney U-test (***$P = 10^{-4}$). **d**, Quantification of the number of anti-FLAG antibodies bound to the same cells before ($1 \times 10^{-4} \pm 7.3 \times 10^{-5}$) and after uncaging ($2.9 \times 10^{-4} \pm 1.9 \times 10^{-4}$), $N = 4$, $n = 19$. Significance was tested using the two-tailed paired Wilcoxon signed-rank test (***$P = 0.001$). **e**, Representative current traces of BK-channel at varying voltage, with the protocol shown in red, in both control and UV-illuminated cells. **f**, Quantification of current density (pA/pF) at 120 mV in control ($37 \pm 20$, $n = 6$) and UV-illuminated cells ($380 \pm 172$, $n = 8$), $N = 4$. Significance was tested using a two-tailed Mann–Whitney U-test (**$P = 0.002$). Data presented as mean ± s.d. Center line in the box plot represents the median, the box represents 25–75% of the data, whiskers represent 1.5 × interquartile range and the square box among data points represents the mean. Scale bars, 10 μm.

**Fig. 4 | Optogenetic release of VRACs. a**, Composition (left) and activation of VRACs (right). **b**, Schematic of VRAC release using PhoCl. VRACs comprising LRRC8A-PhoCl-TMEM and LRRC8E-GFP are expressed in *LRRC8* knockout (KO) cells. Following uncaging, VRACs traffic to the plasma membrane (PM). **c**, Quantification of current density (pA/pF) at −80 mV following 50% hypotonic shock in untransfected HEK293 *LRRC8* knockout cells ($1 \pm 0.5$, $n = 7$, $N = 1$) and in cells transfected with LRRC8A-PhoCl-TMEM and LRRC8E-GFP, without (control) ($1 \pm 0.7$, $n = 12$, $N = 2$) and with UV illumination ($5.6 \pm 4.9$, $n = 10$, $N = 3$). Significance was tested using a two-tailed Mann–Whitney U-test (control versus UV, *$P = 0.03$, untransfected versus UV, *$P = 0.04$). **d**, Current traces of activated VRAC measured using the protocol shown in red in a control and a UV-illuminated cell. **e**, Time course of VRAC currents measured using voltage ramp protocol (shown in red) under isotonic condition, following 50% hypotonic shock and application of DCPIB, in a representative UV-illuminated cell. Data presented as mean ± s.d. Center line in the box plot represents the median, the box represents 25–75% of the data, whiskers represent 1.5 × interquartile range and the square box among data points represents the mean.

in the same cells without 405-nm illumination (Fig. 3b–d). Consistent with this, electrophysiological measurements following uncaging revealed a marked increase in characteristic BK-channel currents compared with control cells (Fig. 3e,f). However, a basal current could be recorded also for control cells that were not illuminated, indicating a minor leakage (Fig. 3e,f). We concluded that our system allows for the controlled release of both functional ion channels and multi-spanning multisubunit transmembrane proteins.

## Reconstitution of VRAC function in knockout cells

Encouraged by these results, we decided to reconstitute an ion channel with more complex regulation. We chose volume-regulated anion channels (VRACs) that are formed by LRRC8 proteins. VRACs open following osmotic cell swelling and exhibit electrophysiological properties defined by their subunit composition[13]. Hexameric VRACs are constituted from at least one LRRC8A subunit that is essential for export from the endoplasmic reticulum and surface delivery and that heteromerizes with at least one LRRC8B-E subunit[14] (Fig. 4a). We expressed LRRC8E-GFP and LRRC8A-PhoCl-TMEM in knockout HEK cells devoid of any VRAC subunit[14] so that ectopically expressed LRRC8 proteins would not reach the cell surface (Fig. 4b). This resulted in bright endo-membrane fluorescence in transfected cells but did not yield VRAC conductance (Fig. 4c). Only when we released functionally assembled VRACs, formed by LRRC8A/E heteromers from the Golgi apparatus by

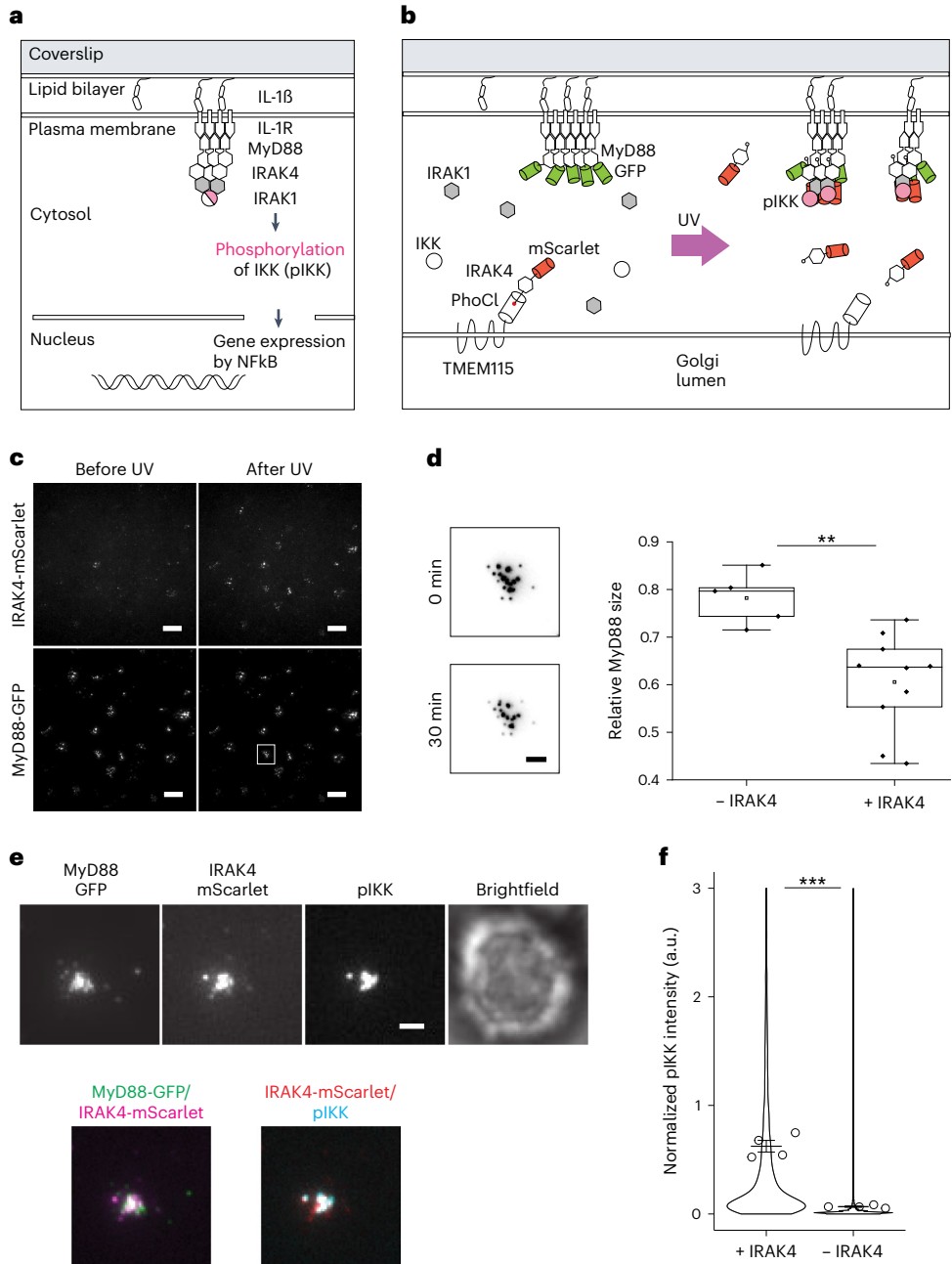

**Fig. 5 | Reconstitution of IL-1R signaling pathway in knockout cells with optogenetically released effector protein. a**, Schematic of IL-1R signaling. **b**, Schematic of TMEM-PhoCl-IRAK4-mScarlet construct in *IRAK4* knockout EL4 cells. In the absence of functional IRAK4 in the cells, IL-1R signaling is hampered and MyD88 forms large clusters. Following release of IRAK4 with UV light, IL-1R signaling resumes and MyD88 cluster size is controlled. IL-1R signaling results in phosphorylation of IKKα/β. **c**, TIRF images of *IRAK4* knockout cells expressing MyD88-GFP and TMEM-PhoCl-IRAK4-mScarlet at 0 and 30 min following UV illumination. Scale bars, 10 μm. *N* = 3. **d**, Left: representative images of the same MyD88-GFP cluster at 0 and 30 min following UV illumination. Scale bar, 2 μm. Right: quantification of relative change in MyD88-GFP cluster size 30 min following UV illumination in MyD88 clusters without IRAK4 recruitment (0.78 ± 0.05, *n* = 5) and in MyD88 clusters with IRAK4 recruitment (0.61 ± 0.10, *n* = 10). Data presented as mean ± s.d. Significance

was tested using the two-tailed unpaired *t*-test (**P = 0.003). *N* = 3. Center line in the box plot represents the median, the box represents 25–75% of the data, whiskers represent 1.5 × interquartile range and the square box among data points represents the mean. **e**, TIRF images showing IRAK4-mScarlet, MyD88-GFP and phosphorylated IKK-AF647 (pIKKα/β) cluster in *IRAK4* knockout cells expressing the TMEM-PhoCl-IRAK4-mScarlet construct following optogenetic release of IRAK4 in the cytosol. Scale bar, 2 μm. **f**, Quantification of pIKKα/β-AF647 intensity in *IRAK4* knockout cells following UV-based IRAK4 release (0.6 ± 0.05, *m* = 2,709) and without IRAK4 release (0.07 ± 0.006, *m* = 112184). Violin plot shows the distribution of individual MyD88 puncta measurements; dots superimposed on the violin plot represent means of replicates (*N* = 4). Bars represent mean ± s.e.m. Significance was tested using the two-tailed unpaired *t*-test (***P = 10⁻⁵), employing the means of replicates as data points. *m*, number of MyD88 puncta analyzed.

UV pulses, did we detect hypotonicity-induced VRAC conductance, but not in cells not illuminated with UV pulses (Fig. 4c,d). Uncaged, plasma membrane-delivered LRRC8A/E-composed VRACs showed the expected kinetics[14], with maximal current observed around 4 min

following hypotonic shock (Supplementary Fig. 5). Furthermore, the recorded currents were completely blocked by 4-(2-butyl-6,7-dichloro-2-cyclopentyl-indan-1-on-5-yl) oxybutyric acid (DCPIB), the most selective inhibitor of VRACs[15,16] (Fig. 4e). We concluded that we

could restore ion channel function, and thus osmotic regulation, in knockout cells by optogenetic release of recombinant VRACs.

### Reconstitution of signaling pathway in knockout cells

Finally, we hypothesized that if a signaling pathway was disrupted by the knockout of an essential protein component of the system, the controlled release of the knocked-out component could restore the signaling event and allow for its quantitative investigation at the single-cell level.

As a model we used a T lymphoblast system in which the interleukin-1 receptor (IL-1R) is activated by its ligand in supported membrane bilayers following cell attachment. In these cells, IL-1R activation leads to the formation of a multiprotein signaling complex referred to as the Myddosome (Fig. 5a). We used a cell line in which (1) Myddosome protein MyD88 was genetically labeled with GFP and (2) *IRAK4*, encoding for an essential kinase in the Myddosome, was knocked out via CRISPR–Cas9 (ref. 17). In these cells, activation of IL-1R did not lead to downstream phosphorylation of the inhibitor of nuclear factor kappa-B kinase (IKK), because of the lack of IRAK4. MyD88 clusters still formed but, in the absence of signaling capacity, became significantly enlarged[17]. When we then expressed IRAK4-mScarlet-PhoCl-TMEM at the Golgi apparatus in these cells and released IRAK4-mScarlet by a short pulse of 405-nm light, we found that IRAK4-mScarlet became recruited to and coassembled with MyD88 puncta on the plasma membrane as detected by TIRF microscopy (Fig. 5b,c). Furthermore, as previously shown[17], MyD88 clusters became smaller, signifying activation for signaling (Fig. 5d). Release of IRAK4 also activated Myddosome complexes leading to signal transduction. Indeed, when we fixed cells following optogenetic release of IRAK4-mScarlet and immunostained them for the phosphorylated form of IKK subunits alpha and beta (pIKKα/β), the downstream output of Myddosome activation, we found that cells containing MyD88 puncta colocalizing with released IRAK4 were also positive for pIKKα/β staining whereas Myddosomes without IRAK4 were not (Fig. 5e,f). We concluded that optogenetic release of knocked-out IRAK4 could restore Myddosome-IKK signaling in single-knockout T lymphoblasts.

## Discussion

We demonstrate here an optogenetic system for the release of single functional molecules in live cells. Our approach overcomes an important bottleneck in the observation of single molecules in that it allows for the controlled release of small amounts of functional protein, compatible with the density requirements of single-molecule imaging. All cytosolic and membrane proteins should be compatible with our system because we have both N- and C-terminal anchors for the Golgi apparatus, and even functional multispanning and multidomain transmembrane complexes such as BK and VRAC ion channels could be optogenetically released. Furthermore our systems achieve quantitative labeling efficiency, thus giving access to single-molecule imaging of any molecule without cellular background that may interfere in the isolated observation of specific function. Our assay worked with uncaging via focused laser, light-emitting diode lamps and UV flashlights and was established in cell lines CHO, HEK, HeLa and CV-1, demonstrating excellent portability of the workflow. We have thus generated a straightforward approach for single-molecule imaging of any membrane or cytosolic protein in live cells.

Optogenetic release of single molecules has the potential to address previously intractable questions regarding the numbers and dynamics of proteins required to execute cellular functions. In this way, the precise molecular function of many proteins could be isolated using a combination of knockout cell lines with quantitative delivery of the depleted molecule via optogenetic release. Here we demonstrate this capability, showing that the optogenetic release of IRAK4 in knockout cells reconstitutes Myddosome formation and IKK activation in IL-1 signaling. In the future our technique may allow determination of the number of activated molecules that are required to reach a tipping point, leading to a functional decision at the level of the entire cell in similar systems.

We demonstrate the reconstitution of specific ion channel conductance in cells otherwise devoid of it (Figs. 3 and 4). The high sensitivity of electrophysiology, as well as our results from single-molecule imaging of released molecules (Fig. 3), demonstrate the extremely low leakiness of our method.

Releasing a controlled wave of functional, tagged proteins in live cells for acute investigation is highly advantageous. In contrast, when a cell expresses a labeled protein of interest in steady state, the population of target proteins could be heterogenous in terms of its conformation, post-translational modifications and interaction with binding partners. In such a situation, any observation of a specific type of process will thus be biased by the pre-existing distribution of states of the molecule. As a result, measurement will be subjected to a high level of noise. Our method, by releasing a small wave of a target molecule, allows for synchronized observation of the sequence of events in which the target molecule is involved as it executes its function. Soluble molecules were observed instantaneously in TIRF microscopy whereas we could detect transmembrane molecules readily 2 h following uncaging. By utilization of TMEM115 and RER, we use Golgi-retained proteins with varying membrane topology as anchors, allowing for N- and C-terminal fusions via PhoCl to serve proteins with varying membrane topology. Because, following photocleavage of PhoCl, a small peptide (11 amino acids) will remain connected to the protein of interest, it should be tested whether this has an impact on protein function.

In the future our method could be further improved in terms of photoconversion efficiency and dissociation rate. PhoCl is one instance of a photocleavable fluorescent protein, and many additional molecules could be engineered to generate future variants. The mechanism by which PhoCl was generated from mMaple[18], circular permutation and mutagenesis should, in principle, be applicable to all fluorescent proteins that undrgo peptide backbone breakage following UV illumination, such as the mEOS proteins[19], Kaede[20], mkikGR[21], Dendra2 (ref. 22) or IrisFP[23]. While we have already used an improved variant of PhoCl[24] with faster release kinetics of the cleaved peptide, we expect that more and further improved photocleavable proteins will be generated in the future, extending the range and capabilities of our concept. Finally, to extend the scope of our method from single-molecule imaging and functional reconstitution in live cells, it may also be possible to release proteins via PhoCl in animals via two-photon-mediated photocleavage, allowing the expression of proteins in specific cells at a specific time within developing embryos or living animals. In this context, precisely timed release of functional protein during organism development or behavior in a knockout background should generate unprecedented resolution of molecular processes in complex, dynamic environments.

## Online content

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

## Methods

### Generation of constructs

For TMEM-PhoCl-FGF2-GFP, complementary DNA encoding for human TMEM115 (Origene, no. RG203956) using NheI and BsrGI restriction sites, PhoCl, amplified from pcDNA-NLS-PhoCl-mCherry (Addgene, no. 8769), using XhoI and HindIII restriction sites, and FGF2-GFP, using EcoRI and SalI restriction sites, were sequentially inserted into vector pEGFP-C1. In all further work described, an improved variant of PhoCl, termed PhoCl-2c[24], was used. Henceforth, PhoCl denotes PhoCl-2c.

For mScarlet-CD4-PhoCl-RER, a fusion construct encompassing a signal peptide (MWPLVAALLLGSACCGSA), genes for mScarlet, the transmembrane region of CD4 (STPVQPMALIVLGGVAGLLLFIGL-GIFFCVRCRHRRR), PhoCl and RER1 were generated by gene synthesis (Thermo Fisher Scientific). The fusion construct was then cloned into vector pTREtight2 (Addgene, no. 19407, a gift from M. Ralser) using NheI and XbaI.

For the mScarlet-CD4-PhoCl-RER construct used in stable cell generation for the UV dose–response assay, fragments encoding the TRE promoter, amplified from vector pTREtight2, and mScarlet-CD4-PhoCl-RER were cloned into AAVS1 Safe Harbor Targeting Knock-in HR Donor 2.0 (System Bioscience, no. GE622A-1) using NEBuilder HiFi DNA Assembly (New England Biolabs) following digestion of the vector with restriction enzymes SpeI and MluI. The guide RNA sequence targeting the AAVS1 safe harbor locus (gRNA sequence: ggggccactagggacaggat) was cloned into vector pSpCas9(BB)-2A-GFP PX458 (Addgene, no. 48138) using the restriction site BbsI. The double-stranded gRNA insert was generated via annealing of two complementary primer oligos with their respective overhangs.

For constructs FLAG-BK-yellow fluorescent protein (YFP)-PhoCl-TMEM and LRRC8A-GFP-PhoCl-TMEM, CACNA1E-eGFP-PhoCl-TMEM115 (in vector pTREtight2) was used as a template from which CACNA1E-eGFP was removed using restriction site AgeI, and fragments for FLAG-BK-YFP and LRRC8A-GFP was inserted, respectively, using Gibson cloning[25,26]. The FLAG-BK-YFP fragment was amplified from plasmid FLAG-BK667YFP[27], a gift from T. Giraldez. Construct LRRC8A-PhoCl-TMEM was generated from plasmid LRRC8A-GFP-PhoCl-TMEM, in which the vector was digested with AgeI and EcoRI and all fragments except GFP were reassembled by Gibson cloning.

For construct TMEM-PhoCl-IRAK4-mScarlet, TMEM-PhoCl including a fragment with a 3xGGS linker was amplified from TMEM-PhoCl-rtTA3-mScarlet (Twist Bioscience). IRAK4, ordered as a gBlock (IDT), was fused to mScarlet via a 3xGGS linker by PCR. A vector backbone was generated using lentiviral plasmid pHR-dSV digested with restriction enzymes Mlu1 and Not1. Construction of the pHR-dSV-TMEM-PhoCl–IRAK4-mScarlet fusion was achieved by Gibson assembly.

All restriction enzymes were purchased from New England Biolabs. Primer sequences are provided in Supplementary Information.

### Cell culture and transfection

CV-1 (ATCC CCL-70) and HeLa-EM2 cells (ATCC, a gift from M. Gossen), used for imaging transmembrane proteins, were maintained with DMEM (Thermo Fisher Scientific, no. 31053028) supplemented with 10% fetal bovine serum (FBS; neoLab, no. 2095ML500) and 1% GlutaMax (Thermo Fisher Scientific, no. 35050061). CHO-K1 cells (ATCC CCL-61, a gift from J. D. Esko) stably expressing TMEM-PhoCl-FGF2-GFP were cultured in minimum essential medium-α (Sigma-Aldrich, no. M8042) supplemented similarly with 10% FBS and 1% GlutaMax. Gene-edited EL4.NOB1 cells (ECACC 87020408, also called EL4 cells) used for Myddosome signaling assays were cultured in RPMI (Thermo Fisher Scientific, no. 32404014) with 10% FBS (Sigma, no. F0804) supplemented with 1% L-glutamine (Thermo Fischer Scientific, no. 25030024). HEK293T cells (Clonetech, no. 632617) used for stable cell line generation were cultured in DMEM (Thermo Fisher Scientific, no. 10938025) supplemented with 2mM L-glutamine, 1 mM sodium pyruvate (Gibco, no. 11360070) and 10% FBS (Sigma, no. F0804). For BK patch-clamp experiments, HEK293 cells (DSMZ, no. ACC-305) were cultured in minimum essential medium (PAN-Biotech, no. P04-09500) supplemented with 6% FBS (PAN-Biotech, no. P04-37500), 1% penicillin/streptomycin (PAN-Biotech, no. P06-07100) and 1% glutamine. For VRAC patch-clamp experiments, LRRC8 knockout HEK293 cells (disrupted of all five *LRRC8* genes)[14] (DSMZ, no. ACC-305, a gift from T. J. Jentsch) were cultured in DMEM (PAN-Biotech, no. P04-03590) supplemented with 10% FBS (PAN-Biotech, no. P30-3302) and 1% glutamate. All cells were maintained at 37 °C with 5% $CO_2$.

Fusion constructs with pTREtight2 vectors were cotransfected using reverse tetracycline-controlled plasmid transactivator 3 (pLenti CMV rtTA3 Hygro (w785-1); Addgene, no. 26730, a gift from E. Campeau) in the presence of doxycycline (Clontech, no. 631311). pLenti CMV rtTA3 Hygro is henceforth referred to as rtTA3.

### Generation of stable cell line

CHO-K1 cells expressing TMEM-PhoCl-FGF2-GFP in a doxycycline-dependent manner were established via retroviral transduction using a Moloney murine leukemia virus-based system. Recombinant virus particles were produced in HEK 2-293 cells (Clontech, no. 631507) transfected with TMEM-PhoCl-FGF2-GFP in vector pRev-TRE2, containing a TET-responsive element. Transfection and virus production utilized the MBS Mammalian Transfection Kit (Agilent Technologies). The infectious supernatant was filtered onto CHO-K1 cells, leading to stable expression of MCAT-1 and rtTA2-M2. Subsequent fluorescent activated cell sorting (FACS), including a step without doxycycline, ensured the selection of positive cell populations.

For the generation of EL4 cells stably expressing TMEM-PhoCl-IRAK4-mScarlet, cotransfection of pHR transfer plasmids with second-generation packaging plasmids pMD2.G and psPAX2 (a gift from D. Trono; Addgene plasmid nos. 12259 and 12260) was done to produce lentivirus in HEK293T cells. After 72 h, harvested supernatant was filtered and applied to *IRAK4* knockout EL4 cells. Cells were then subjected to FACS for selection of populations positive for pHR-dSV-TMEM-PhoCl-IRAK4-mScarlet. EL4.NOB1 cells were used as a negative control for gating of FACS. Cells were sorted using a BD FACS Aria II at the Deutsches Rheuma-Forschungszentrum Berlin Flow Cytometry Core Facility.

For the UV dose–response PhoCl activation assay, a stable cell line expressing mScarlet-CD4-PhoCl-RER was generated. HeLa-EM2 cells stably expressing rtTA2 (ref. 28) were transfected with a combination of pSpCas9(BB)-2A-GFP PX458 and AAVS1 Safe Harbor Targeting Knock-in HR Donor 2.0 in a 1:1 ratio using Pei. Fourteen days following transfection, expression of mScarlet-CD4-PhoCl-RER was induced with doxycycline (0.04 µg ml$^{-1}$) and positive cells were sorted via FACS (BD FACS ARIA III Cellsorter).

### Live-cell microscopy

PhoCl activation was performed on an inverted Olympus IX71 microscope equipped with a Yokogawa CSU-X1 spinning disk; a ×60/1.42 numerical aperture Olympus oil objective was used, together with lasers at 491 nm (100 mW, Cobolt) and 561 nm (100 mW; Cobolt). A quad-edge dichroic beam splitter (446/523/600/677 nm; Semrock) was used to separate fluorescence emission from excitation light, and final images were taken with an Orca Flash 4.0 sCMOS camera (Hamamatsu). Images were acquired with MetaMorph (Molecular Devices).

All TIRF single-molecule-tracking experiments were performed on a custom-built microscope[29]. An Olympus TIRF objective (×60/1.49 numerical aperture) was used with a 473-nm laser (100 mW; Laserglow Technologies) and a 643-nm laser (150 mW; Toptica Photonics). A quad-edge dichroic beam splitter (405/488/561/635 nm; Semrock) separated fluorescence emission from excitation light. Emission light was further filtered by a quad-band bandpass filter (446/523/600/677 nm; Semrock) and focused by a 500-mm tube lens onto the chip of a back-illuminated,

electron-multiplying, charge-coupled device camera (Evolve, Photometrics) that was water-cooled to −85 °C. Images were acquired with MicroManager 2.0 (ref. 30).

## Uncaged protein single-molecule TIRF assay

CHO-K1 cells expressing TMEM-PhoCl-FGF2-GFP were illuminated with 10-s pulses of a 405-nm laser at 2 mW mm$^{-2}$ for uncaging. Videos were acquired before and after uncaging in TIRF mode. The number of molecules on the plasma membrane was counted using the Trackmate[31] plugin in ImageJ[32] and divided by cell surface area.

CV-1 cells were transfected with plasmids mScarlet-CD4-PhoCl-RER and rtTA3 using the Neon Transfection System (Thermo Fisher Scientific) and seeded onto glass-bottom, 35-mm gridded dishes (Ibidi, no. 81168) in the presence of doxycycline (0.02 µg ml$^{-1}$). Cells were incubated with red fluorescent protein (RFP) antibody (Chromotek, no. 5f8-2) self-labeled with AF647 (dye:antibody ratio, 15:1, concentration = 16 µg ml$^{-1}$) and imaged by TIRF microscopy before and 2 h after uncaging. For the uncaging process, cells were illuminated five times with 5-s pulses of a 405-nm laser at 33 mW mm$^{-2}$ every 15 s for. Pulse-chase assay for BK-channels was performed similarly. FLAG-BK-YFP-PhoCl-TMEM was cotransfected with rtTA3 in the presence of doxycycline (1 µg ml$^{-1}$). FLAG antibody (Sigma-Aldrich, no. F1804, 1:800 dilution) and secondary antibody labeled with AF647 (Invitrogen, no. A31571, 1:800 dilution) were used to detect the secreted BK-channels 6–8 h following the uncaging step. The number of molecules on the plasma membrane was counted similarly using ImageJ. All box plots and statistical tests were carried out using OriginLab.

## UV dose–response PhoCl activation assay

Stable cells expressing mScarlet-CD4-PhoCl-RER were induced with doxycycline (0.04 µg ml$^{-1}$) and seeded 24 h before the experiment. Cells were exposed to variable UV dosage (5, 9 and 14 mW mm$^{-2}$) for 15 s using an arc lamp (Newport, model OPS-A1000) with a 400-nm filter followed by incubation for 3 h at 37 °C. Cells were then either fixed for microscopy or collected for flow cytometry.

Cells were fixed with a solution comprising 4% paraformaldehyde, 0.2% glutaraldehyde and 10% FBS prepared in PBS, quenched and incubated with RFP antibody self-labeled with AF647 (dye:antibody ratio, 2:1, concentration = 5 µg ml$^{-1}$) before imaging by TIRF microscopy.

For the preparation of cells for flow cytometry assay, these were treated with cold solutions and kept on ice throughout. Cells were incubated with Versene (Thermo Fisher Scientific) for 10 min and then scraped and collected. They were washed with PBS and then incubated with RFP antibody self-labeled with AF647 (dye:antibody ratio, 11:1, concentration = 3 µg/ml$^{-1}$) for 30 min. Cells were washed with PBS before being resuspended in PBS with 5% FBS. Flow cytometry measurements were performed with a BD FACSCanto II analyzer and analysis performed using FlowJo. The gating strategy used in the flow cytometry experiments is detailed in Supplementary Fig. 4.

## IRAK4 reconstitution assay

Imaging of EL4 cells was performed on an inverted microscope (Nikon TiE) equipped with a Nikon fiber launch TIRF illuminator. Illumination was controlled with a laser combiner using laser lines of 488, 561 and 640 nm. Fluorescence emission was collected with a Nikon Plan Apo ×100/1.4 numerical aperture oil immersion objective and filters for GFP (525 ± 25 nm), RFP (595 ± 25 nm) and AF647 (700 ± 75 nm) and projected onto a Photometrics 95B Prime sCMOS camera with 2 × 2 binning (calculated pixel size of 150 nm) and a ×1.5 magnifying lens. Image acquisition was performed using NIS-Elements software. Live-cell experiments were performed at 37 °C in an OKO Labs heated microscope enclosure. EL4 cells expressing MyD88-GFP and IRAK4-mScarlet-PhoCl-TMEM were illuminated with a 405-nm laser (21 mW mm$^{-2}$) for 10 s. TIRF images were acquired post uncaging for 30 min at 30-s intervals.

## IL-1β-functionalized SLBs

For the creation of supported lipid bilayers (SLBs), phospholipid mixtures consisting of 97.5% mol 1-palmitoyl-2-oleoyl-*sn*-glycero-3-phosphocholine, 2% mol 1,2-dioleoyl-sn-glycero-3-[(*N*-(5-amino-1-carboxypentyl)iminodiacetic acid)succinyl] (ammonium salt) (DGS-NTA) and 0.5% mol 1,2-dioleoyl-sn-glycero-3-phosphoethanolamine-*N*-[methoxy(polyethylene glycol)-5000] were mixed in glass, round-bottom flasks and dried down with a rotary evaporator. All lipids were purchased from Avanti Polar Lipids. Dried lipids were placed under vacuum for 2 h to remove trace chloroform and resuspended in PBS. Small unilamellar vesicles (SUVs) were produced by freeze–thaw cycles and bath sonication. Once the suspension had cleared, SUVs were spun in a benchtop ultracentrifuge at 35,000g for 45 min. SUVs were stored at 4 °C for up to 1 week.

For preparation of SLBs on 96-well, glass-bottom plates, these were cleaned for 30 min with 5% Hellmanex solution containing 10% isopropanol heated to 50 °C then incubated with 5% Hellmanex solution for 1 h at 50 °C, followed by washing with distilled water. Plates were then dried with nitrogen gas and sealed until needed. Individual wells were removed and their bases etched for 15 min with 5 M KOH, then washed with PBS. SUV suspension was deposited in each well and allowed to form for 1 h at 45 °C. Following washing with PBS, SLBs were incubated for 15 min with HEPES buffered saline (HBS: 20 mM HEPES, 135 mM NaCl, 4 mM KCl, 10 mM glucose, 1 mM CaCl$_2$, 0.5 mM MgCl$_2$) and 10 mM NiCl$_2$ to charge the DGS-NTA lipid with nickel. SLBs were then washed in HBS containing 0.1% bovine serum albumin (BSA) to block the surface and minimize nonspecific protein adsorption. Following blocking, SLBs were functionalized by incubation with His10-IL-1β for 1 h. The labeling solution was washed out and SLBs further functionalized with His10-Halo-IL-1β for 1 h. Excessive ligands were washed out with HBS.

## Immunofluorescence staining

For analysis of the activation of phospho-IKKα/β on MyD88-GFP puncta, cells were stimulated with IL-1β-functionalized SLBs for 30 min followed by 405-nm illumination to release PhoCl-IRAK4. After 45 min, cells were fixed with 3.5% paraformaldehyde containing 0.5% Triton X-100 for 20 min at room temperature. Following blocking with PBS and 10% BSA at 4 °C overnight, cells were incubated with antiphospho-IKKα/β (Cell Signaling, no. 2697, 1:400) and Nano-secondary AF647 (Chromotek, no. srbAF647-1-100, 1:1,000) prepared in PBS with 10% BSA and 0.1% Triton X-100 for 1 h at room temperature. Cells were labeled with FluoTag-X4 anti-GFP conjugated to Atto488 (NanoTag Biotechnologies, no. N0304-At488-L, 1:500) and FluoTag-X2 anti-mScarlet conjugated to Atto565 (NanoTag Biotechnologies, no. N1302-At565-L, 1:500) and imaged with TIRF microscopy.

## Analysis of MyD88-GFP puncta and pIKKα/β staining

For quantification of the intensity of MyD88-GFP puncta, images were processed in ImageJ to remove background fluorescence using custom-written macros[17]. Postprocessing steps included image subtraction with a dark-frame image to remove camera noise, followed by image subtraction with a median-filtered image to remove background associated with cytosolic fluorescence. The dark-frame image was acquired with no light exposure to the camera but with exposure time identical to that of experimental acquisition, while the median-filtered image was generated in Fiji with a radius of 25 pixels. Mean intensity from individual MyD88-GFP puncta was measured. The intensity of MyD88 puncta 30 min following UV illumination was divided by intensity at 0 min to estimate the relative change in MyD88 cluster size following IRAK4 uncaging.

Total internal reflection fluorescence microscopy images of pIKKα/β immunofluorescence staining were quantified using an established pipeline[33]. Dark-frame and median-filtered images were subtracted from MyD88-GFP- and pIKKα/β-staining TIRF micrographs. MyD88-GFP puncta were segmented and their fluorescence intensity

measured using a custom Cell Profiler pipeline. The integrated intensity and mean intensity of MyD88, IRAK4 and pIKKα/β were extracted for each segmented punctum. Data processing and intensity normalization were performed using the following equation with R:

Norm.Int = (Intensity − quantile(0.01))/(quantile(0.99) − quantile(0.01)).

MyD88 puncta were defined as IRAK4 positive when IRAK4 normalized integrated intensity was ≥0.5. pIKKα/β normalized integrated intensity was plotted.

## BK-channel patch-clamp assay
HEK293 cells transfected with FLAG-BK-YFP-PhoCl-TMEM and rtTA3 were seeded in the presence of doxycycline (1 µg ml$^{-1}$). After 24 h, cells were illuminated with 15-s pulses of UV at 1.42 mW mm$^{-2}$ using an UV lamp (Uvico-RAPP Optoeletronic) and whole-cell patch-clamp recordings were performed at room temperature 7 h following uncaging. Control measurements were simultaneously performed on separate dishes.

Currents were acquired using an Axopatch 200B amplifier and the Axograph acquisition program (Axograph Scientific) via an Instrutech ITC-18 D-A interface (HEKA Elektronik). Currents were filtered at 5 kHz and digitized at 20 kHz. The bath solution comprised the following: 145 mM NaCl, 5 mM KCl, 1 mM MgCl$_2$, 10 mM HEPES and 3 mM ethylene glycol-bis(β-aminoethyl ether)-N,N,N′,N′-tetraacetic acid (EGTA), pH 7.4. Given the high conductance of BK-channels, the following internal (pipette) solution was used to reduce current and minimize series resistance error: 90 mM NMDG-Cl, 50 mM KCl, 4 mM NaCl, 10 mM HEPES, 1 mM Mg-ATP, 10 mM EGTA and 2 mM CaCl$_2$, pH 7.3 (refs. 34,35). Pipette resistance was 2–6 MΩ and cell capacitance 3–25 pF, as measured by the compensating circuit of the amplifier. We accepted a maximal voltage-clamp error (calculated as the product of maximum current and uncompensated series resistance) of 10 mV. The standard IV protocol used to elicit BK currents consisted of 100-ms voltage steps ranging from −80 to +140 mV in 20-mV increments, starting from a holding potential of 0 mV. No leak-current subtraction was performed. Figures were prepared using Igor Pro (https://www.wavemetrics.com/products/igorpro).

## VRAC patch-clamp assay
*LRRC8* knockout HEK293 cells (disrupted of all five *LRRC8* genes)[14] were seeded in the presence of doxycycline (1 µg ml$^{-1}$) and transfected with LRRC8E-GFP, LRRC8A-PhoCl-TMEM and rtTA3 using the Ca$_3$(PO$_4$)$_2$ technique. For uncaging, cells were illuminated with 10-s pulses of 405-nm light (58 mW mm$^{-2}$) (CoolLED pE-300ultra multiband spectrum). Experiments were carried out 4 h following UV exposure while control measurements were simultaneously performed on separate dishes.

Whole-cell voltage-clamp experiments were performed in isotonic extracellular solution containing 150 mM NaCl, 6 mM KCl, 1 mM MgCl$_2$, 1.5 mM CaCl$_2$, 10 mM glucose and 10 mM HEPES, pH 7.4 with NaOH (320 mOsm). VRAC currents were elicited by perfusion of cells with hypotonic solution containing 75 mM NaCl, 6 mM KCl, 1 mM MgCl$_2$, 1.5 mM CaCl$_2$, 10 mM glucose and 10 mM HEPES, pH 7.4 with NaOH (160 mOsm). The pipette solution contained 40 mM CsCl, 100 mM Cs-methanesulfonate, 1 mM MgCl$_2$, 1.9 mM CaCl$_2$, 5 mM EGTA, 4 mM Na$_2$ATP and 10 mM HEPES, pH 7.2 with CsOH (290 mOsm). Osmolarities of all solutions were assessed with an Osmometer OM 807 freezing-point osmometer (Vogel). All experiments were performed at a constant temperature of 20–22 °C. Currents were recorded with an EPC-10 USB patch-clamp amplifier and PatchMaster software (HEKA Elektronik). Patch pipettes had a resistance of 3–5 MΩ. Currents were sampled at 5 kHz and low-pass filtered at 10 kHz. The holding potential was −30 mV. The standard protocol for measurement of the time course of VRAC current activation consisted of a 0.6-s step to −80 mV followed by a 2.6-s ramp from −100 to 100 mV, which was applied every

12 s. Readout for VRAC current was steady-state, whole-cell current at −80 mV normalized to cell capacitance (current density) subtracted by baseline current density at −80 mV before the application of hypotonic solution. The voltage protocol, applied before the standard protocol and following complete activation of VRAC, consisted of 2-s steps from −120 to 80 mV, with 20-mV increments, preceded and followed by a 0.5-s step to −80 mV every 5 s. The voltage-step protocol confirmed VRAC-typical properties of outward rectification and depolarization-dependent inactivation for LRRC8A/E-containing VRACs[14]. In uncaged cells, VRAC currents were blocked by 100 µM DCPIB (Tocris).

## Reporting summary
Further information on research design is available in the Nature Portfolio Reporting Summary linked to this article.

## Data availability
Microscopy and electrophysiology data are available on request from the corresponding author. Source data are provided with this paper.

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

## Acknowledgements

We thank T. Giraldez for the kind gift of the plasmid encoding the BK-channel Slo1, T. J. Jentsch for *LRRC8* knockout cells and M. Gossen for HeLa-EM2 cells. This work was funded by Deutsche Forschungsgemeinschaft (DFG, German Research Foundation) as part of TRR/SFB186 (project no. 278001972, to H.E. and W.N.) and under Germany's Excellence Strategy (no. EXC-2049-390688087, 'NeuroCure'). A.J.R.P. is a recipient of a Heisenberg Professorship from DFG (project no. 446182550). Work in the laboratory of R.E.C. was supported by grants from the Canadian Institutes of Health Research (no. FS-154310) and the Natural Sciences and Engineering Research Council of Canada (no. RGPIN-2018-04364). Work in the laboratory of M.J.T. was supported by the Max Planck Society and DFG (project no. 499533619). We thank A. Klemmer, C. Knappe, D. Shyshko, A. Senge and A. Elhazaz Fernandez for help with molecular biology,

and all members of the Ewers laboratory for helpful discussions. We acknowledge the assistance of Y. Weber and the Core Facility for Flow Cytometry at the Center for Infectious Diseases FU Berlin.

## Author contributions

H.E. and P.K. conceived of and designed the project. P.K., X.L., R.E.C., F.B., N.A.S., C.S.-L., W.N. and R.S. contributed reagents. D.B. helped in the initial stages of the project. P.K. and F.B. performed and analyzed all single-molecule-imaging experiments. C.S.-L. and P.K. designed, performed and analyzed flow cytometry experiments. S.B. and A.J.R.P. designed, performed and analyzed the electrophysiology experiments for BK-channels. Y.K. and T.S. designed, performed and analyzed the electrophysiology experiments for VRAC. F.C., N.A.S. and M.J.T. designed, performed and analyzed the experiments with IRAK4. H.E. and P.K. wrote the paper, with input from all authors. S.B., F.C., Y.K. and F.B. contributed equally and A.J.R.P., T.S. and M.J.T. contributed equally.

## Funding

## Competing interests

The authors declare no competing interests.

## Additional information

**Correspondence and requests for materials** should be addressed to Helge Ewers.

# Reporting Summary

## Statistics

For all statistical analyses, confirm that the following items are present in the figure legend, table legend, main text, or Methods section.

| n/a | Confirmed | |
|---|---|---|
| ☐ | ☒ | The exact sample size (*n*) for each experimental group/condition, given as a discrete number and unit of measurement |
| ☐ | ☒ | A statement on whether measurements were taken from distinct samples or whether the same sample was measured repeatedly |
| ☐ | ☒ | The statistical test(s) used AND whether they are one- or two-sided<br>*Only common tests should be described solely by name; describe more complex techniques in the Methods section.* |
| ☒ | ☐ | A description of all covariates tested |
| ☒ | ☐ | A description of any assumptions or corrections, such as tests of normality and adjustment for multiple comparisons |
| ☐ | ☒ | A full description of the statistical parameters including central tendency (e.g. means) or other basic estimates (e.g. regression coefficient) AND variation (e.g. standard deviation) or associated estimates of uncertainty (e.g. confidence intervals) |
| ☐ | ☒ | For null hypothesis testing, the test statistic (e.g. *F*, *t*, *r*) with confidence intervals, effect sizes, degrees of freedom and *P* value noted<br>*Give P values as exact values whenever suitable.* |
| ☒ | ☐ | For Bayesian analysis, information on the choice of priors and Markov chain Monte Carlo settings |
| ☒ | ☐ | For hierarchical and complex designs, identification of the appropriate level for tests and full reporting of outcomes |
| ☒ | ☐ | Estimates of effect sizes (e.g. Cohen's *d*, Pearson's *r*), indicating how they were calculated |

*Our web collection on statistics for biologists contains articles on many of the points above.*

## Software and code

Policy information about availability of computer code

| Data collection | Metamorph 7.8.6.0, Micromanager 2.0, NIS-elements 5.11.01, Axograph Scientific 1.7, Patch Master v2x91 |
|---|---|
| Data analysis | Fiji 1.54f, Cell Profiler 4.2.1, IgorPro 9.0, OriginLab 2022b, R 4.2.2, FlowJo v10.10.0 |

For manuscripts utilizing custom algorithms or software that are central to the research but not yet described in published literature, software must be made available to editors and reviewers. We strongly encourage code deposition in a community repository (e.g. GitHub). See the Nature Portfolio guidelines for submitting code & software for further information.

## Data

Policy information about availability of data

All manuscripts must include a data availability statement. This statement should provide the following information, where applicable:
- Accession codes, unique identifiers, or web links for publicly available datasets
- A description of any restrictions on data availability
- For clinical datasets or third party data, please ensure that the statement adheres to our policy

Source data has been provided in excel files. Original images and current traces are too big to upload and are available upon request from the corresponding author.

## Research involving human participants, their data, or biological material

Policy information about studies with human participants or human data. See also policy information about sex, gender (identity/presentation), and sexual orientation and race, ethnicity and racism.

| | |
|---|---|
| Reporting on sex and gender | N/A |
| Reporting on race, ethnicity, or other socially relevant groupings | N/A |
| Population characteristics | N/A |
| Recruitment | N/A |
| Ethics oversight | N/A |

Note that full information on the approval of the study protocol must also be provided in the manuscript.

# Field-specific reporting

Please select the one below that is the best fit for your research. If you are not sure, read the appropriate sections before making your selection.

☒ Life sciences          ☐ Behavioural & social sciences          ☐ Ecological, evolutionary & environmental sciences

For a reference copy of the document with all sections, see nature.com/documents/nr-reporting-summary-flat.pdf

# Life sciences study design

All studies must disclose on these points even when the disclosure is negative.

| | |
|---|---|
| Sample size | The sample size varied depending on the type of experiments and is given in the figure legends for all experiments contained in this study. No pre study sample size calculations were performed. The sample size was determined based on the experimental design, the duration of the full experiment in a day and the time required for imaging live samples without stressing them. The electrophysiology experiments were limited by the number of successful patch clamps possible per day. The arbitrarily chosen sample size was sufficient to show significant difference. |
| Data exclusions | To restrict the analysis to fluorescent puncta that were bonafide Myddosomes, we perform the following normalization procedure. The intensities of MyD88, IRAK4 and pIKK at the 1% quantile were used as the minimum for normalization. We normalized the intensities using the equation: Norm. Int = (Intensity - quantile(0.01))/(quantile(0.99) - quantile(0.01)). We then rounded normalized intensities to 2 decimals. If puncta had negative intensities (intensity value < 0) after normalisation, they were removed from the analysis, to ensure the analysis of pIKK staining was restricted to segmented puncta that most likely were true Myddosome signaling complexes. To confirm this analysis did not possibly exclude true Myddosomes, we performed manual inspection of the segmented images and found this procedure only resulted 1 segmented punctum out of 114894 puncta being excluded from further data analysis visualization. For data visualization purpose, the violin plots only show pIKK normalized intensity between 0-3, value > 3 were not shown on the plots but were included in calculating the mean. |
| Replication | Biological replicates varied depending on the type of experiments and is given in the figure legends for all experiments contained in this study. |
| Randomization | There are two categories in the study: control and UV treated samples. Cells were randomly assigned a category before application of UV, analysed and plotted for the final experiments through automated pipeline. Electrophysiology experiments, due to technical limitations (use of LED or lamps for UV illumination) were performed such that whole dishes were considered either illuminated or not illuminated (control samples). In this setup randomization of samples was not possible. |
| Blinding | All cells in predetermined control and UV treated groups were pooled together, analysed and plotted in an automated pipeline. |

# Reporting for specific materials, systems and methods

We require information from authors about some types of materials, experimental systems and methods used in many studies. Here, indicate whether each material, system or method listed is relevant to your study. If you are not sure if a list item applies to your research, read the appropriate section before selecting a response.

## Materials & experimental systems

| n/a | Involved in the study |
|---|---|
| ☐ | ☒ Antibodies |
| ☐ | ☒ Eukaryotic cell lines |
| ☒ | ☐ Palaeontology and archaeology |
| ☒ | ☐ Animals and other organisms |
| ☒ | ☐ Clinical data |
| ☒ | ☐ Dual use research of concern |
| ☒ | ☐ Plants |

## Methods

| n/a | Involved in the study |
|---|---|
| ☒ | ☐ ChIP-seq |
| ☐ | ☒ Flow cytometry |
| ☒ | ☐ MRI-based neuroimaging |

# Antibodies

| Antibodies used | anti-RFP (supplier: Chromotek, clone nr.: 5F8, lot nr.: 11041)<br>anti-FLAG (supplier: Sigma-Aldrich, product nr.: F1804, batch nr.: SLCK5688)<br>anti-phospho-IKK (Supplier: Cell Signaling Technology, #2697)<br>Nano-secondary AF647 (Chromotek, #srbAF647-1-100)<br>FluoTag-X4 anti-GFP conjugated to Atto488 (NanoTag Biotechnologies, #N0304-At488-L)<br>FluoTag-X2 anti-mScarlet-i conjugated to Atto565 (NanoTag Biotechnologies, #N1302-At565-L) |
|---|---|
| Validation | anti-RFP (supplier: Chromotek, clone nr.: 5F8, lot nr.: 11041): Antibody was tested against CD4mRFP expressing live cells in TIRF microscopy in the lab. In the company website, immunofluorescence images of cells expressing MannosidaseII-tdTomato (Golgi) show good colocalisation with the antibody staining.<br><br>anti-FLAG (supplier: Sigma-Aldrich, product nr.: F1804, batch nr.: SLCK5688): Antibody was tested against Flag-BK expressing live cells in TIRF microscopy in the lab. The company website cites https://doi.org/10.1186/s12985-016-0610-7 for the use of the antibody for immunofluorescence where cells expressing glycoproteins gD-flag/gM-flag were costained with anti-FLAG.<br><br>anti-phospho-IKK (Supplier: Cell Signaling Technology, #2697): The antibody has been validated for WB, immunohistochemistry, and Flow cytometry by the company. For example, Flow cytometry example shows clear difference between control and, TPA and LPS treated THP cells using the antibody.<br><br>FluoTag-X4 anti-GFP conjugated to Atto488 (NanoTag Biotechnologies, #N0304-At488-L): Company website show multiple examples images of immunostaining. Examples include cells expressing Nup98-GFP (nuclear pore protein, colocalization with GFP) and TOM70-nfGFP-BFP(mitochondria and colocalization with BFP).<br><br>FluoTag-X2 anti-mScarlet-i conjugated to Atto565 (NanoTag Biotechnologies, #N1302-At565-L): Company website show multiple example images of immunostaining. Examples include cells expressing mScarlet-i-tubulin (microtubules, colocalization with mScarlet) and TOM70-nfmScarlet-BFP (mitochondria). |

# Eukaryotic cell lines

Policy information about cell lines and Sex and Gender in Research

| Cell line source(s) | CV1 from ATCC<br>HEK293T from Deutsche Sammlung von Mikroorganismen und Zellkulturen, Germany<br>CHO-K1 from ATCC<br>EL-4 from Clontech<br>HeLa from ATCC<br>HeLa-EM2 expressing rtTA2-M2 were a gift from Manfred Gossen laboratory. They are modified HeLa cells from ATCC. |
|---|---|
| Authentication | Cell lines were not authenticated |
| Mycoplasma contamination | Cells were tested negative for mycoplasma. |
| Commonly misidentified lines<br>(See ICLAC register) | Not used |

# Flow Cytometry

## Plots

Confirm that:

☒ The axis labels state the marker and fluorochrome used (e.g. CD4-FITC).

☒ The axis scales are clearly visible. Include numbers along axes only for bottom left plot of group (a 'group' is an analysis of identical markers).

☒ All plots are contour plots with outliers or pseudocolor plots.

☒ A numerical value for number of cells or percentage (with statistics) is provided.

## Methodology

| | |
|---|---|
| Sample preparation | Described in Methods. Briefly, cells were detached using Versene, scraped and collected. The cells were washed multiple times with PBS before and after antibody treatment. They were centrifuged and the supernatant was removed for the washing process. The cells were resuspended in PBS with 5 % FBS prior to measurements |
| Instrument | BD FACSCanto II analyser |
| Software | BD FACSDiva Software, FlowJo |
| Cell population abundance | Final cell population post gating varied between 0-90% |
| Gating strategy | Live cells were selected from SSC-A vs FSC-A plots. From the live cell population, single cells were selected from FSC-H vs FSC-A plots. mScarlet/ RFP positive population was selected according to the negative non expressing control using the SSC-A vs 561 nm plots. AF647 labelled antibody binding population was selected according to the negative control expressing the construct but not uncaged via UV light using the SSC-A vs 647 nm plots. |

☒ Tick this box to confirm that a figure exemplifying the gating strategy is provided in the Supplementary Information.

