## [Peer Review File · Nature Methods]

Peer Review Information

Manuscript Title: An optogenetic method for the controlled release of single molecules

Corresponding author name(s): Helge Ewers

Editorial Notes: None

Reviewer Comments & Decisions:

Decision Letter, initial version:

Dear Helge,

Please let me begin by apologizing for the long time your paper spent in review. Upon receiving the reviews, we had a brief back and forth with the refs, which took a little extra time as well.

Your Article, "An optogenetic method for the controlled release of single molecules", has now been seen by three reviewers. As you will see from their comments below, although the reviewers find your work of considerable potential interest, they have raised a number of concerns. We are interested in the possibility of publishing your paper in Nature Methods, but would like to consider your response to these concerns before we reach a final decision on publication.

We therefore invite you to revise your manuscript to address these concerns. We ask that you focus on addressing the technical questions raised by refs 2 and 3. Regarding the concerns of novelty, please emphasize that in the original PhoCl paper, only overexpressed proteins were used and it was not shown that they could be released at levels suitable for single-molecule studies.

* include a point-by-point response to the reviewers and to any editorial suggestions

* please underline/highlight any additions to the text or areas with other significant changes to facilitate review of the revised manuscript

- * address the points listed described below to conform to our open science requirements
- * ensure it complies with our general format requirements as set out in our guide to authors at www.nature.com/naturemethods
- * resubmit all the necessary files electronically by using the link below to access your home page

[Redacted] This URL links to your confidential home page and associated information about manuscripts you may have submitted, or that you are reviewing for us. If you wish to forward this email to co-authors, please delete the link to your homepage.

We hope to receive your revised paper within three months. If you cannot send it within this time, please let us know. In this event, we will still be happy to reconsider your paper at a later date so long as nothing similar has been accepted for publication at Nature Methods or published elsewhere.

OPEN SCIENCE REQUIREMENTS

REPORTING SUMMARY AND EDITORIAL POLICY CHECKLISTS

IMAGE INTEGRITY

- that unprocessed scans are clearly labelled and match the gels and western blots presented in

figures.

- that control panels for gels and western blots are appropriately described as loading on sample processing controls
- all images in the paper are checked for duplication of panels and for splicing of gel lanes.

DATA AVAILABILITY

All novel DNA and RNA sequencing data, protein sequences, genetic polymorphisms, linked genotype and phenotype data, gene expression data, macromolecular structures, and proteomics data must be deposited in a publicly accessible database, and accession codes and associated hyperlinks must be provided in the "Data Availability" section.

MATERIALS AVAILABILITY

As a condition of publication in Nature Methods, authors are required to make unique materials

promptly available to others without undue qualifications.

SUPPLEMENTARY PROTOCOL

To help facilitate reproducibility and uptake of your method, we ask you to prepare a step-by-step Supplementary Protocol for the method described in this paper. We [encourage authors to share their step-by-step experimental protocols](https://www.nature.com/nature-research/editorial-policies/reporting-standards#protocols) on a protocol sharing platform of their choice and report the protocol DOI in the reference list. Nature Portfolio 's Protocol Exchange is a free-to-use and open resource for protocols; protocols deposited in Protocol Exchange are citable and can be linked from the published article. More details can found at www.nature.com/protocolexchange/about.

ORCID

Sincerely,
Rita

Rita Strack, Ph.D.
Senior Editor
Nature Methods

Reviewers' Comments:

Reviewer #1:

Remarks to the Author:

Kashyap et al have leveraged a previously described optogenetic system PhoCl to control the biological function of several proteins, mediated by UV irradiation. PhoCl is a photocleavable protein engineered by Campbell et al (Nat. Methods 2017, 14, 391). The proteins whose activity have been sequestered were a cytosolic protein (EGFP) and single-pass transmembrane domain (CD4) and a subunit of an ionic channel. The localization and biological function of the mentioned proteins were restored after irradiation with UV light.

On the one hand, I would like to congratulate the authors for the experiments done, which I believe were properly designed and have the right controls to support the claimed conclusions. In this case, the controls only require the omission of UV light, as it is needed to release the target protein to restore its activity.

On the other hand, I have my doubts about the novelty and scientific impact of the paper. In the original paper published by Campbell et al, besides doing the engineering of the photocleavable features of PhoCl, they also showed how it is possible to control the biological activity of a soluble protein by sequestering through a transmembrane domain to other organelles of the cell. For instance, they controlled the location of a fluorescent protein between the cytoplasm and the nucleus (Figure 2), the enzymatic activity of a protease and finally they were able to restore the function of the ion channel Panx1 (Figure 3). In the title and the text, the authors mentioned that this technique can be used for imaging by super resolution microscopy, as a small number of molecules can be released. However, in the manuscript I do not see any experiment done using super-resolution microscope, single-molecule biophysics nor any biological discovery associated to those techniques.

I would like to mention that the quality of the images is very good, and the presentation of the experiments clearly done. Also, the references are appropriate and support the conclusions.

Minor issues:

The references 23 and 24 are the same.

Reviewer #2:

Remarks to the Author:

- A. Summary of the key results
- B. Originality and significance: if not novel, please include reference
- C. Data & methodology: validity of approach, quality of data, quality of presentation
- D. Appropriate use of statistics and treatment of uncertainties
- E. Conclusions: robustness, validity, reliability
- F. Suggested improvements: experiments, data for possible revision
- G. References: appropriate credit to previous work?
- H. Clarity and context: lucidity of abstract/summary, appropriateness of abstract, introduction and conclusions

A. +

B. Kashyap et al. present a system for optogenetic release of single molecules in live cells. Their method allows for the insertion of functional proteins in the plasma membrane upon UV light pulse triggered release from the Golgi. They test their method for simple transmembrane proteins, ion channels and soluble intracellular signaling proteins (a kinase involved in immune cell signaling). They convincingly show that these molecules are functional, and that reconstitution is at single molecule levels.

I don't know of any other study that uses photo-release from the Golgi to achieve "expression" at the single-molecule level. Like the authors say, this is actually a difficulty when expressing fluorescently labels proteins for single-molecule imaging: more often than not, expression levels are too high for single molecule analysis. Given the convincing functional data, I do believe that this method could be very useful for the single-molecule imaging community, but probably also for functional assays in which cellular function is restored after knock-out (and that don't have to be studied by single-molecule methods per se; combining retention in the Golgi and photo release is still a neat idea).

C. All in all, I find the data shown convincing and well presented. Some comments on the data and presentation:

- 1) In all cases, I can see some mScarlet signal in the membrane also before UV radiation. What type of background is this? Noise? Or after all some spurious release from the Golgi in the absence of UV irradiation?
- 2) Figure 4D: I find it very strange to use a box plot representation for normalized data. I would encourage the authors to show the absolute intensities here and maybe stick to the type of plot used for similar data in Figures 1 and 2, e.g. Figure 2D. Also, for the same figure, it should be explained that assembled MyDDosomes can be seen (but are not functional) without IRAK-4, and that the shrinking of the MyDDosomes is a way to also test for functional IRAK-4 recruitment.
- 3) Figure 1J: unit on x axis is missing. Can the authors comment on the non-significant correlation and the outliers (the cells for which the amount of surface detected CD4 is seemingly too low?) Do they have any explanation for this? A technical one?

D. Fine. See also C. point 3 for one question about statistics.

E. While I find the conclusions well justified by the presented data, I really miss a more technical discussion for a methods paper. Like this, I cannot judge how cumbersome this method is to use, and this would of course make a big difference in how readily it is adopted.

- 1) How easy is / was it to set these experiments up? Could this be done in any cell (with Golgi), or could there be limitations?
- 2) Can the authors comment on how easy it was to adapt the pulsing conditions to each studied protein? How much tweaking was required? In other words: how easy or straightforward is it to apply this method to new targets?
- 3) For some of the experiments, there was a long delay between uncaging and detection of the release molecule (up to 7 hours). I missed a discussion about this in the discussion section.
- 4) I noticed that the authors use different Golgi "anchors". Can you explain why you picked TREM115 and RER and how the two differ? It would be nice to have some guidance about this for future users.

F. None, I note this is also already a revised version with additional data that supports the usefulness of the method well.

G. To my knowledge, yes.

H. See E. for additional points worthwhile to add in the discussion/conclusion section. Also, one small limitation of the methods is that N- and C-terminal tagging are not symmetric, as in one case the beta barrel of PhoCl remains bound to the molecule of interest. Can the authors comments on how much influence they expect this to have on protein behaviour?

Minor comments:

p.2, l. 8ff "the controlled delivery of single molecules remains an unsolved challenge as existing approaches are either too leaky for control at the single molecule level since they are based on noncovalent attachment or require complex expression systems such as mRNA injection." Can the authors provide reference(s) for these claims? I would also point out that "leaky" promoters are actually a useful tool to achieve single-molecule compatible expression levels!

p.2, l. 15ff I had to read this several times to understand that the point was that PhoCl remains bound to the Golgi protein rather than the protein of interest if C-terminally attached. I think the authors could just directly say that in the text.

p.4, l.6ff I would prefer some unpolished, not perfectly quantitative data in the SI to "data not shown" here – even if the variation of the uncaging pulse was not done systematically, there should be some data to support the claim.

Reviewer #3:

Remarks to the Author:

In this paper, the authors show with great elegance how it is possible to confine a transmembrane protein to the Golgi Apparatus with the help of a linking protein. The linking protein is cleavable by an optogenetic element and a small light dose. This technique, the authors suggest as a controlled delivery method for single molecules.

Comments:

1. The authors write on page 2 line 21 – 23: "At the same time the fusion to a Golgi-resident protein allowed for covalent attachment to the Golgi apparatus, resulting in low leakiness."

An explanation of leakiness would be appropriate here. And with references to other papers with the use of the term.

2. The supplementary movies are very beautiful. For the sake of analyzing the single particles in the movie which is also suggested by the authors, it would be beneficial to bring down the particle density. How would the authors do this. The Golgi is very crowded and if all proteins are labelled it will be difficult to assess single trajectories from single molecules if the density is too high in the acquired movies.

3. The authors claim that: p6 line 23 "the controlled release of the knocked-out component could restore the signaling event and allow for its quantitative investigation at the single cell level." I agree and think that this is the most important argument for why this technical development pushes forward

the field of single molecule imaging. This is mentioned in the discussion but it could be even more clear when concluding the paper. I would also suggest to mention this in the title of the paper.

Author Rebuttal to Initial comments

We thank the reviewers for their productive input and their helpful suggestions. Please find below our answers to all points raised by the reviewer. We added additional experimental data in Figure 2 in the main text and supplementary data. We furthermore made changes in the text to address all points raised by the reviewers. We hope our manuscript in its improved form is found to be acceptable for publication by the reviewers.

Reviewers' Comments:

Reviewer #1:

Remarks to the Author:

Kashyap et al have leveraged a previously described optogenetic system PhoCI to control the biological function of several proteins, mediated by UV irradiation. PhoCI is a photocleavable protein engineered by Campbell et al (Nat. Methods 2017, 14, 391). The proteins whose activity have been sequestered were a cytosolic protein (EGFP) and single-pass transmembrane domain (CD4) and a subunit of an ionic channel. The localization and biological function of the mentioned proteins were restored after irradiation with UV light.

On the one hand, I would like to congratulate the authors for the experiments done, which I believe were properly designed and have the right controls to support the claimed conclusions.

In this case, the controls only require the omission of UV light, as it is needed to release the target protein to restore its activity.

On the other hand, I have my doubts about the novelty and scientific impact of the paper. In the original paper published by Campbell et al, besides doing the engineering of the photocleavable features of PhoCI, they also showed how it is possible to control the biological activity of a soluble protein by sequestering through a transmembrane domain to other organelles of the cell. For instance, they controlled the location of a fluorescent protein between the cytoplasm and the nucleus (Figure 2), the enzymatic activity of a protease and finally they were able to restore the function of the ion channel Panx1 (Figure 3). In the title and the text, the authors mentioned that this technique can be used for imaging by super resolution microscopy, as a small number of molecules can be released. However, in the manuscript I do not see any experiment done using super-resolution microscope, single-molecule biophysics nor any biological discovery associated to those techniques.

We thank the reviewer for their assessment, with which we respectfully disagree of course. While the abovementioned work suggests a number of possible future applications, and shows some relocation data, this was all done with strongly overexpressed protein and no work in knockout cells reconstituting specific functions nor single molecule work was done. In fact, as we state in our manuscript, we failed for months using the original PhoCI to do the work described here. It was only after using the updated version of this exciting new optogenetic tool, PhoCI 2.0, that we succeeded in reliably releasing single molecules.

This allowed us to follow the motion of single transmembrane molecules on the plasma membrane, which in itself is an accomplishment that we as single molecule biologists are extremely excited about. Furthermore we then could, by releasing molecules optogenetically, restore the function of ion channels and of immune signaling in cells, in which the respective function was disrupted by knockout. We could thus reconstitute essential functions by adding back single components in a tightly controlled manner. This was not possible in the previous work. We are convinced this is a breakthrough in the investigation of the function of multiprotein assemblies in live cells. Also, we now clearly

show that the release can be controlled in a dose-dependent manner, both by FACS and single molecule microscopy (!) which is in our view an extremely convincing proof of function and general applicability.

I would like to mention that the quality of the images is very good, and the presentation of the experiments clearly done. Also, the references are appropriate and support the conclusions.

Minor issues:

The references 23 and 24 are the same.

We apologize for the mistake, we now addressed this issue.

Reviewer #2:

Remarks to the Author:

- A. Summary of the key results
- B. Originality and significance: if not novel, please include reference
- C. Data & methodology: validity of approach, quality of data, quality of presentation
- D. Appropriate use of statistics and treatment of uncertainties
- E. Conclusions: robustness, validity, reliability
- F. Suggested improvements: experiments, data for possible revision
- G. References: appropriate credit to previous work?
- H. Clarity and context: lucidity of abstract/summary, appropriateness of abstract, introduction and conclusions

A. +

B. Kashyap et al. present a system for optogenetic release of single molecules in live cells. Their method allows for the insertion of functional proteins in the plasma membrane upon UV light pulse triggered release from the Golgi. They test their method for simple transmembrane proteins, ion channels and soluble intracellular signaling proteins (a kinase involved in immune cell signaling). They convincingly show that these molecules are functional, and that reconstitution is at single molecule levels.

I don't know of any other study that uses photo-release from the Golgi to achieve "expression" at the single-molecule level. Like the authors say, this is actually a difficulty when expressing fluorescently labeled proteins for single-molecule imaging: more often than not, expression levels are too high for single molecule analysis. Given the convincing functional data, I do believe that this method could be very useful for the single-molecule imaging community, but probably also for functional assays in which cellular function is restored after knock-out (and that don't have to be studied by single-molecule methods per se; combining retention in the Golgi and photo release is still a neat idea).

We thank the reviewer for this assessment of our work.

C. All in all, I find the data shown convincing and well presented. Some comments on the data and presentation:

1) In all cases, I can see some mScarlet signal in the membrane also before UV radiation. What type of background is this? Noise? Or after all some spurious release from the Golgi in the absence of UV irradiation?

We thank the reviewer for this question, indeed, as often, there are several sources of background signal. In this experiment to include only surface exposed mScarlet of successfully released molecules, we use detection of mScarlet via anti-mScarlet antibodies coupled to AF647. Sources of detected points before UV-induction may thus be: 1) unspecific binding of these antibodies, 2) possible leakage of individual molecules to the plasma membrane in the absence of cleavage and 3) bright signal due to "dirt", some organic material in the sample.

2) Figure 4D: I find it very strange to use a box plot representation for normalized data. I would encourage the authors to show the absolute intensities here and maybe stick to the type of plot used for similar data in Figures 1 and 2, e.g. Figure 2D.

We thank the reviewer for pointing this out. This has been changed.

Also, for the same figure, it should be explained that assembled MyDDosomes can be seen (but are not functional) without IRAK-4, and that the shinking of the MyDDosomes is a way to also test for functional IRAK-4 recruitment.

We thank the reviewer for pointing this out. We now explain this in the text.

3) Figure 1J: unit on x axis is missing.

Can the authors comment on the non-significant correlation and the outliers (the cells for which the amount of surface detected CD4 is seemingly too low?) Do they have any explanation for this? A technical one?

We thank the reviewer for pointing this out. We now replaced Figure 1J with new Figure 2d,e, which shows dose-response experiments for optogenetic release as superior form of quantification.

D. Fine. See also C. point 3 for one question about statistics.

E. While I find the conclusions well justified by the presented data, I really miss a more technical discussion for a methods paper. Like this, I cannot judge how cumbersome this method is to use, and this would of course make a big difference in how readily it is adopted.

1) How easy is / was it to set these experiments up? Could this be done in any cell (with Golgi), or could there be limitations?

We thank the reviewer for this important question. We performed systematic search for optimal UV pulses and optimized for power and pulse length. After we found the conditions reported in the manuscript for FGF2-GFP and CD4-mScarlet, our system was straight forward to adapt to the other systems and worked even for uncaging not with a focused 405 nm laser beam, but also UV flash lamps and LED lights. We performed experiments in a variety of cell lines (HeLa, HEK, CV-1, CHO) and think this should work in any cell (with a Golgi, if Golgi sequestering is used). We now discuss this also in the text.

2) Can the authors comment on how easy it was to adapt the pulsing conditions to each studied protein? How much tweaking was required? In other words: how easy or straightforward is it to apply this method to new targets?

We thank the reviewer for this question. As mentioned above, we established the system *de novo* on a number of vastly different constructs (C-terminal PhoCl, N-terminal PhoCl, several PhoCl anchors, single PhoCl anchor, soluble protein, multispinning transmembrane protein). Also, we established it in 4 different laboratories with 4 different cell lines, so we are confident to say that the method is straight-forward to apply to new targets.

3) For some of the experiments, there was a long delay between uncaging and detection of the release molecule (up to 7 hours). I missed a discussion about this in the discussion section.

We thank the reviewer for this question. To make sure to detect ion channel current, we started our experiments at 12h post uncaging moving progressively forward after successful experiments. We reached 2 hours after uncaging for transmembrane proteins consistent with the biosynthetic trafficking pathway. The soluble FGF2 could be detected instantaneously after uncaging. We now mention this in the discussion.

4) I noticed that the authors use different Golgi “anchors”. Can you explain why you picked TREM115 and RER and how the two differ? It would be nice to have some guidance about this for future users.

We apologize for the omission and thank the reviewer for making us aware. By using in TMEM115 and RER, we use Golgi-retained transmembrane proteins with different membrane topology as anchors, allowing for N- and C-terminal fusions via PhoCI to serve proteins with different membrane topologies.

F. None, I note this is also already a revised version with additional data that supports the usefulness of the method well.

G. To my knowledge, yes.

H. See E. for additional points worthwhile to add in the discussion/conclusion section. Also, one small limitation of the methods is that N- and C-terminal tagging are not symmetric, as in one case the beta barrel of PhoCI remains bound to the molecule of interest. Can the authors comments on how much influence they expect this to have on protein behaviour?

We thank the reviewer for this question. Generally, with PhoCI being a fluorescent protein, we expect any effect to be similar that expected for any fluorescent protein tag. The interference of a tag the size of a fluorescent protein has on protein function has to be determined empirically for each case. For the VRAC-channel for example, a fluorescent protein cannot be attached to the N-terminus, as it would interfere with function. This is why it is important we offer constructs both for C-terminal and N-terminal linkage. We did not observe any functional problems in the cases we used. However, we never observed any ER membrane aggregates as expected for artefactual dimerization of fluorescent proteins. For small proteins there may be a relatively strong influence on diffusion if the 20 kDa of the PhoCI shell makes up a significant part of the entire construct mass.

Minor comments:

p.2, l. 8ff “the controlled delivery of single molecules remains an unsolved challenge as existing approaches are either too leaky for control at the single molecule level since they are based on noncovalent attachment or require complex expression systems such as mRNA injection.” Can the authors provide reference(s) for these claims? I would also point out that “leaky” promoters are actually a useful tool to achieve single-molecule compatible expression levels!

We thank the reviewer for this comment. One reference that discusses the need for low level expression is Max Ulbrichs review on “Counting Molecules: Toward Quantitative Imaging” that carefully discusses a number of caveats in single molecule imaging, including the requirement for low level expression: https://doi.org/10.1007/4243_2011_36

Indeed, several solutions have been developed to allow for low level expression and leaky promoters have been successfully used. However, we present here a precisely temporally inducible release of a small amount of molecules in a dose response controlled manner (new Figure 2d,e), which leaky promoters do not offer.

p.2, l. 15ff I had to read this several times to understand that the point was that PhoCI remains bound to the Golgi protein rather than the protein of interest if C-terminally attached. I think the authors could just directly say that in the text.

We thank the reviewer for pointing this out, indeed this was poorly worded, we now specifically inserted the sentence: “UV-induced cleavage of PhoCI would thus result in the barrel structure of PhoCI to remain at the Golgi apparatus, whereas the C-terminal peptide and the fused target molecule would be released” into the text at the appropriate site.

p.4, l.6ff I would prefer some unpolished, not perfectly quantitative data in the SI to “data not shown” here – even if the variation of the uncaging pulse was not done systematically, there should be some data to support the claim. We thank the reviewer for this comment. We now added a series of experiments, in which we could show using single molecule imaging and FACS analysis that the release of molecules increased with the intensity of the light pulse used for uncaging (new Figure 2d,e).

Reviewer #3:

Remarks to the Author:

In this paper, the authors show with great elegance how it is possible to confine a transmembrane protein to the Golgi Apparatus with the help of a linking protein. The linking protein is cleavable by an optogenetic element and a small light dose. This technique, the authors suggest as a controlled delivery method for single molecules.

Comments:

1. The authors write on page 2 line 21 – 23: “At the same time the fusion to a Golgi-resident protein allowed for covalent attachment to the Golgi apparatus, resulting in low leakiness.”

An explanation of leakiness would be appropriate here. And with references to other papers with the use of the term. We thank the reviewer for this question. Leaky as a term in this context was used to point out the non-covalent nature of other caging methods, resulting in unintended release of molecules before the induced release. Our system is inspired from the very elegant and effective RUSH system. However due to the tethering being based on streptavidin-biotin bond when the hook and the POI are binding, it is prone to a premature loss of molecules that prohibits single molecule imaging in many cases (for example not stoichiometric expression). The term is used for “leaky promoters” for example that in their off state have low level transcription.

2. The supplementary movies are very beautiful. For the sake of analyzing the single particles in the movie which is also suggested by the authors, it would be beneficial to bring down the particle density. How would the authors do this. The Golgi is very crowded and if all proteins are labelled it will be difficult to assess single trajectories from single molecules if the density is too high in the acquired movies.

We thank the reviewer for this question. Indeed, the particle density in Supplementary Movie 2 is too high for single particle tracking, albeit the individual spots are single molecules. To control the amount of particles overall released to the plasma membrane for unambiguous single particle tracking, we can adjust laser power. We now show a dose-response analysis of the release of CD4-mScarlet molecules to the plasma membrane in dependence of UV power both by single molecule tracking as well as FACS analysis (Figure 2).

3. The authors claim that: p6 line 23 “the controlled release of the knocked-out component could restore the signaling event and allow for its quantitative investigation at the single cell level.” I agree and think that this is the most important argument for why this technical development pushes forward the field of single molecule imaging. This is mentioned in the discussion but it could be even more clear when concluding the paper. I would also suggest to mention this in the title of the paper.

We thank the reviewer for this interesting suggestion, we will discuss it with the editor.

Decision Letter, first revision:

Dear Helge,

Thank you for submitting your revised manuscript "An optogenetic method for the controlled release of single molecules" (N METH-A51538C). It has now been seen by the original referees and their comments are below. The reviewers find that the paper has improved in revision, and therefore we'll be happy in principle to publish it in Nature Methods, pending minor revisions to comply with our editorial and formatting guidelines.

We are now performing detailed checks on your paper and will send you a checklist detailing our editorial and formatting requirements early next year following the holidays. Please do not upload the final materials and make any revisions until you receive this additional information from us.

TRANSPARENT PEER REVIEW

ORCID

Sincerely,
Rita

Rita Strack, Ph.D.

Senior Editor
Nature Methods

Reviewer #1 (Remarks to the Author):

The authors have submitted a revised version of the original manuscript. When I was reading the paper, I specifically looked to see if the authors were using the most evolved version, PhoCl2.0 developed by Campbell et al (Chem. Sci., 2021, 12, 9658). That is why I noticed that the references to this paper were duplicated.

On one hand, the authors have leveraged a method previously developed to decrease the number of photons to control the amount of protein released. On the other hand, the release, translocation, and recovery of a biological activity through a beta-elimination reaction of a photocleavable protein has already been reported (Nature Methods 2017 14, 391–394 & Chem. Sci., 2021, 12, 9658). Therefore, to my understanding, from the protein engineering and optogenetic point of view, I do not see any novelty.

I would like to highlight that the experiments and controls were very well done, the quality of the images is excellent, and the conclusions are reasonable.

However, I cannot comment on the importance of this technical advancement (not method) and the usefulness for single-molecule biophysicists or cell biologists. Hence, I humbly leave the decision to the editorial team.

Reviewer #2 (Remarks to the Author):

I think that the edits the authors made in response to the reviewers' comments have significantly improved the paper. They have addressed all my comments on the previous version.

Author Rebuttal, first revision:

Response to reviewers

Reviewer #1:

Remarks to the Author:

The authors have submitted a revised version of the original manuscript. When I was reading the paper, I specifically looked to see if the authors were using the most evolved version, PhoCl2.0 developed by Campbell et al (Chem. Sci., 2021, 12, 9658). That is why I noticed that the references to this paper were duplicated.

On one hand, the authors have leveraged a method previously developed to decrease the number of photons to control the amount of protein released. On the other hand, the release, translocation, and recovery of a biological activity through a beta-elimination reaction of a photocleavable protein has already been reported (Nature Methods 2017 14, 391–394 & Chem. Sci., 2021, 12, 9658). Therefore, to my understanding, from the protein engineering and optogenetic point of view, I do not see any novelty.

I would like to highlight that the experiments and controls were very well done, the quality of the images is excellent, and the conclusions are reasonable.

However, I cannot comment on the importance of this technical advancement (not method) and the usefulness for single-molecule biophysicists or cell biologists. Hence, I humbly leave the decision to the editorial team.

We thank the reviewer for their comment. We agree that we did not do anything new in terms of protein engineering (except developing a system to confine proteins to the Golgi apparatus via their N-or C-termini). We of course disagree though that the method for the release of single molecules is not new. What we present here, releasing a small, controlled amount of molecules to restore a function at the level of few molecules was not possible before (and not done in the original PhoCl paper). We believe that we provide a significant advance in that while our paper was one of the first building on PhoCl, there will be many building on what we made possible. As single molecule microscopy is a key technique to understand the function of proteins in live cells, we firmly believe our work will open new doors in research. We hope we could convince the reviewer and thank them for including the last two sentences in their review.

Reviewer #2:

Remarks to the Author:

I think that the edits the authors made in response to the reviewers' comments have significantly improved the paper. They have addressed all my comments on the previous version.

Reviewer #3:

None

Final Decision Letter:

Dear Helge,

I am pleased to inform you that your Article, "An optogenetic method for the controlled release of single molecules", has now been accepted for publication in Nature Methods. The received and accepted dates will be August 9, 2023 and Feb 1, 2024. This note is intended to let you know what to expect from us over the next month or so, and to let you know where to address any further questions.

Over the next few weeks, your paper will be copyedited to ensure that it conforms to Nature Methods style. Once your paper is typeset, you will receive an email with a link to choose the appropriate publishing options for your paper and our Author Services team will be in touch regarding any additional information that may be required. It is extremely important that you let us know now whether you will be difficult to contact over the next month. If this is the case, we ask that you send us the contact information (email, phone and fax) of someone who will be able to check the proofs and deal with any last-minute problems.

Please note that *Nature Methods* is a Transformative Journal (TJ). Authors may publish their research with us through the traditional subscription access route or make their paper immediately open access through payment of an article-processing charge (APC). Authors will not be required to make a final decision about access to their article until it has been accepted. [Find out more about Transformative Journals](https://www.springernature.com/gp/open-research/transformative-journals)

Authors may need to take specific actions to achieve [compliance with funder and institutional open access mandates](https://www.springernature.com/gp/open-research/funding/policy-compliance-faqs). If your research is supported by a funder that requires immediate open access (e.g. according to [Plan S principles](https://www.springernature.com/gp/open-research/plan-s-compliance)) then you should select the gold OA route, and we will direct you to the compliant route where possible. For authors selecting the subscription publication route, the journal's standard licensing terms will need to be accepted, including [self-archiving policies](https://www.springernature.com/gp/open-research/policies/journal-policies). Those licensing terms will supersede any other terms that the author or any third party may assert apply to any version of the manuscript.

If you have posted a preprint on any preprint server, please ensure that the preprint details are updated with a publication reference, including the DOI and a URL to the published version of the

article on the journal website.

If you are active on Twitter/X, please e-mail me your and your coauthors' handles so that we may tag you when the paper is published.

Best regards,
Rita

Rita Strack, Ph.D.
Senior Editor
Nature Methods